# The Associations between Depression and Sugar Consumption Are Mediated by Emotional Eating and Craving Control in Multi-Ethnic Young Adults

**DOI:** 10.3390/healthcare12191944

**Published:** 2024-09-28

**Authors:** Austin J. Graybeal, Jon Stavres, Sydney H. Swafford, Abby T. Compton, Stephanie McCoy, Holly Huye, Tanner Thorsen, Megan E. Renna

**Affiliations:** 1School of Kinesiology and Nutrition, University of Southern Mississippi, Hattiesburg, MS 39406, USA; jonathon.stavres@usm.edu (J.S.); sydney.swafford@usm.edu (S.H.S.); abby.compton@usm.edu (A.T.C.); stephanie.mccoy@usm.edu (S.M.); holly.huye@usm.edu (H.H.); tanner.thorsen@usm.edu (T.T.); 2School of Psychology, University of Southern Mississippi, Hattiesburg, MS 39406, USA; megan.renna@usm.edu

**Keywords:** depression, emotion regulation, sugar, dietary intake, eating behaviors, cravings

## Abstract

Background/Objectives: Individuals with mental health conditions such as depression are vulnerable to poor dietary habits, potentially due to the maladaptive eating behaviors often used to regulate negative emotion. However, the specific dietary components most associated with depression, as well as the mediating roles of emotion regulation and other eating behaviors, remains ambiguous in young adults. Methods: For this cross-sectional evaluation, a total of 151 (86 F, 65 M; BMI: 22.0 ± 5.1 kg/m^2^; age: 21.4 ± 2.5 y) multi-ethnic participants (50 White, 36 Black, 60 Asian, and 5 White Hispanic) completed a digital 24-h dietary recall and self-reported measures of depressive symptoms, emotional regulation, and eating behaviors. LASSO regression was used to identify the dietary variables most associated with each subscale and to remove extraneous dietary variables, and multiple regression and mediation analyses were conducted for the remaining variables. Results: Out of >100 dietary factors included, only added sugar in the combined sample (*p* = 0.043), and relative sugar in females (*p* = 0.045), were retained and positively associated with depressive symptoms. However, the relationships between depression and added and relative sugar intake were mediated by craving control and emotional eating, respectively. Individuals with higher added sugar intake (*p* = 0.012–0.037), and females with higher relative sugar intake (*p* = 0.029–0.033), had significantly higher odds of risk for major depression disorder and the use of mental health medications. Conclusions: Added and relative sugar intake are significantly associated with depressive symptoms in young adults, but these relationships may be mediated by facets of emotional dysregulation, such as emotional eating and craving control.

## 1. Introduction

Depression is associated with increased negative emotionality and greater difficulty in regulating negative emotions [1,2]. Emotional regulation, defined as the process by which individuals change the type, frequency, or intensity of an emotional response [3], can sometimes refer to an internal (i.e., cognitive) process; though externalizing (i.e., behavioral) emotional regulation is also common amongst people with and without psychopathologies such as depression. Behavioral attempts at regulating one’s emotions often include increased substance use [4], engagement in self-harm behaviors [5], and dysregulated eating behaviors [6,7]. Specifically, individuals experiencing depressive symptoms are more prone to emotional- and over-eating as a means of self-regulation, and individuals that struggle to control their eating are more prone to depressive symptoms. Notably, the link between eating behavior and depression is clinically relevant, evident by the inclusion of dysfunctional appetite as a symptom of major depressive disorder (MDD) in the DSM-5.

Clearly, mental health is highly vulnerable to maladaptive dietary habits, which is further supported by the link between dietary intake and depressive symptoms [8]. For instance, studies show that women with greater depression demonstrate poorer collective dietary habits when compared to their counterparts [9]. The bidirectionality of this relationship suggests that absence of essential nutrients and the overconsumption of ultra-processed low-nutrient foods may increase an individual’s susceptibility to depressive symptoms, or that depression increases an individual’s susceptibility to the under- and overconsumption of essential and low-nutrient foods, respectively [10]. Given that people tend to eat combinations of foods rather than single nutrients/foods, it is unsurprising that nutrition and mental health professionals have traditionally emphasized strategies that address these collective dietary habits. However, because comprehensive dietary alterations and analyses are exceedingly difficult and have low success rates in the modern food environment, identifying specific dietary components most associated with depressive symptoms has become increasingly popular. The surge in interest is likely due to the belief that identifying and addressing the intake of specific nutrients/foods is more feasible; where a more straightforward approach may improve depressive symptoms without overwhelming dietary changes and may help clinicians more easily identify depression risk via dysregulated eating behaviors. Nevertheless, these specific dietary components remain unclear, and our current understanding of the relationship between individual nutrients and depression is subject to several pitfalls.

First, studies often evaluate the link between diet and depression by independently examining a nutrient of interest without considering the overlap/collinearity with other dietary factors [11]. For example, total carbohydrate intake has shown to be positively associated with depression [12], whereas studies examining carbohydrate sub-classes [13,14,15], or vitamins/minerals derived from carbohydrate sources [16], have revealed distinctly different relationships. The concerns with single nutrient analyses are highlighted by recent studies, where machine learning models reveal that >20 dietary components are uniquely associated with depressive symptoms [17]. Moreover, it is common for investigations examining this relationship to include >10–20 covariates into predictive models, resulting in overfitting errors that reduce the likelihood of replication in external samples [18,19]. This may explain the mixed evidence despite decades of investigation. Young adults are particularly affected, as the links between diet and depression were established using age groups unreflective of the relationships amongst young adults in the modern food environment. Finally, few investigations have accounted for the mediating role of emotional dysregulation, which is more susceptible to dysfunction in young adults [20].

Therefore, the purpose of this study was to evaluate the associations among depressive symptoms, emotional regulation, eating behaviors, and dietary intakes in a multi-ethnic sample of young adults. We hypothesized that there would be positive associations among depression, emotional regulation, and dysregulated eating behaviors, and that rigorous variable selection procedures would identify unique associations between dietary and psychiatric components. We further hypothesized that the dietary intakes retained following absolute shrinkage and selection operator (LASSO) regression procedures would be significantly associated with depression, but that the relationship between depression and diet would be mediated by components of emotional regulation and eating behavior. A second aim of this study was to assess whether specific dietary intakes would be associated with an increased risk of major depression and mental health medication use among the young adults in our sample. To that end, we hypothesized that the dietary factors retained in our depression models would be associated with an increased risk of both major depression disorder (MDD) and the use of prescription mental health medications.

## 2. Materials and Methods

### 2.1. Participants and Design

A total of 171 participants between the ages of 18 and 39 were prospectively recruited for this cross-sectional evaluation through a combination of convenience and snowball sampling (i.e., in-person and online word of mouth). Participants were excluded if they were younger than 18 or older than 39; were pregnant; planning to become pregnant; or breastfeeding or lactating. Of the 171 participants recruited, 15 were excluded due to scheduling conflicts and 5 were excluded due to complete non-response on at least one questionnaire. Thus, 151 multi-ethnic (50 White, 36 Black, 60 Asian, and 5 White Hispanic) male and female (86 F, 65 M) young adults were included in the final analysis (BMI: 22.0 ± 5.1 kg/m^2^; age: 21.4 ± 2.5 y). The study protocol was approved by the university Institutional Review Board and conducted in accordance with the Declaration of Helsinki. All participants provided written informed consent. This study was prospectively registered at ClinicalTrials.gov (NCT05885672) and is ancillary to, but separate from, a larger line of investigation. All primary outcome variables from the present study are distinct from, but were collected in addition to, those stemming from the larger line of investigation.

### 2.2. Procedures

Participants arrived at the laboratory between the hours of 600–1000 after an ≥8-h overnight fast from food, beverages, supplements and medication, and after abstaining from planned exercise for ≥24-h. Upon arrival, participants completed a demographic and health history questionnaire and subsequently underwent measurements of height using a stadiometer, weight using a calibrated digital scale (SECA, Hamburg, Germany), and estimates of body composition using dual-energy X-ray absorptiometry ([DXA] Lunar iDXA v18 enCORE software, General Electric, Boston, MA, USA). For DXA, participants were positioned according to the recommended guidelines, which included reflection scanning for larger participants [21]. Following all health history and anthropometric assessments, participants were escorted to a private room and asked to complete an automated digital 24-h dietary recall and several digital questionnaires (Survey Tool, Qualtrics^®^ LLC, Provo, UT, USA) that asked them about their depressive symptoms, eating behaviors, and emotional regulation.

### 2.3. 24-Hour Dietary Recall

Participants were required to complete a 24-h dietary recall using the web-based Automated Self-Administered 24-Hour Dietary Assessment Tool (ASA24^®^; version#: ASA24-2024). Because this study’s procedures were conducted in accordance with a larger project, only one 24-h dietary recall was collected; however, the ASA24^®^ was developed by the US National Cancer Institute (Bethesda, MD, USA) using the multiple-pass method employed by the national surveillance program [22], and has well-demonstrated performance when compared to actual/usual dietary intake and interviewer-administered recalls [23,24]. Participants were instructed to report all the foods and beverages they consumed over the preceding 24-h, which were used to automate their individual dietary intakes. For testing, participants were asked to assign each reported food item to a respective meal type and report the time that the meal was consumed. After all meals were reported, participants were asked to provide additional information regarding how each meal/food item was prepared and how much was eaten; which was further assessed by prompting participants to choose between visual images of varying portion sizes. All automated dietary intake data, other than the data for water and alcohol consumption (*n* = 2), were extracted from the ASA24^®^, resulting in 101 dietary variables that were used during the variable selection procedures described hereafter. All macronutrients and their respective subcategories were measured in absolute (grams) and relative intakes (% of total energy).

### 2.4. Dietary Intake Variable Selection

All 101 dietary variables automated by the ASA24^®^ were considered for this study as this represents the amount of dietary information provided to clinicians and researchers from many dietary analysis programs. Due to the large number of dietary variables produced by the ASA24^®^ and the potential for considerable multicollinearity between these variables [11], LASSO regression procedures were used to determine the strongest dietary correlates of depression, eating behavior, and emotional regulation subscales. For example, the emotional eating subscale from the Three Factor Eating Questionnaire (TFEQ) was included as the dependent variable (DV), and all dietary variables were included into a single model as potential predictor variables (independent variable [IV]) using LASSO. LASSO regression then works by imposing restrictions on the IV that shrink their coefficients towards 0 (the same as removing the variable from the model entirely) [25], ultimately generating the most parsimonious models by retaining variables avoidant of prediction error and removing unnecessary variables. Simply, LASSO regression penalizes unnecessary variables/variables with significant multicollinearity by shrinking their coefficients until they no longer have an impact on a given model as opposed to eliminating them outright. The λ value used to determine the LASSO shrinkage technique was identified using 10-fold cross-validation and the 1-SE rule [26].

Although the dietary variables were included as IVs during LASSO, the retained dietary variables were included as DV for multiple regression analyses. For example, if total dairy intake was retained as a correlate of craving control following LASSO, craving control was entered as the IV in the subsequent multiple regression model to determine its association with the DV, total dairy, given that psychiatric variables likely predict dietary intakes in the context of this study as opposed to the alternative. Importantly, LASSO regression procedures do not produce inferential statistics (i.e., do not provide *p*-values). Instead, it is simply a systematic approach for the removal of variables that contribute to elevated multicollinearity during regression procedures. In addition, the standardized coefficients and inferential statistics (i.e., *p*-value) produced for a pair of variables during simple multivariate (i.e., including model covariates) regression are mathematically equivalent irrespective of their assignment as an IV or DV (i.e., if an IV and DV are swapped, the standardized coefficient and *p*-value are the same for the switched variables). When mediators are included into multiple regression models, the *p*-values produced for a given IV and DV are mathematically equivalent irrespective of their designation as an IV or DV. As such, using the dietary intakes as the IVs for LASSO regression did not impact their associations/interpretations in subsequent multiple regression models. Moreover, this allowed us to also remove multicollinearity among these variables while also limiting the expected increases in type I error rate if each of the >100 dietary variables were included as a DV in their own multiple regression model.

### 2.5. Depression

The Center for Epidemiologic Studies Depression Scale (CESD) was used to provide data on the participants’ depressive symptoms. The CESD is a 20-item scale that asks participants to indicate the frequency at which they experienced depressive symptoms over the past week [27]. CESD scores range from 0–60 with higher scores indicating greater depressive symptoms. A score of ≥16 on the CESD is consistent with clinically significant depressive symptoms that align with a diagnosis of MDD [27]. As such, participants with a score of ≥16 were classified as being at risk for MDD.

### 2.6. Emotion Regulation

The Emotion Regulation Questionnaire (ERQ) is a validated 10-item measure of adaptive and maladaptive emotional regulation skills [28]. The measure is comprised of two distinct subscales: cognitive reappraisal and expressive suppression. Cognitive reappraisal refers to the adjustment of one’s emotional response by reinterpreting the situation that prompted the emotion (i.e., changing one’s thoughts), whereas expressive suppression refers to the intentional or unintentional inhibition or repression of an emotional response [29]. An example question for the cognitive reappraisal subscale includes “When I want to feel more positive emotion, I change what I am thinking about”, and an example question for expressive suppression includes “I keep my emotions to myself” [28]. Participants respond to these items on a 7-point Likert scale ranging from strongly disagree to strongly agree; with higher scores reflecting higher use of each emotion regulation skill [28].

### 2.7. Eating Behavior

The validated 18-item TFEQ was used to assess typical eating behaviors [30]. The TFEQ includes subscales of: emotional eating (EmE), defined as eating in response to negative emotional cues; uncontrolled eating, defined as a combination of subjective hunger and loss of control that results in excessive eating; and cognitive restraint, which refers to the conscious restriction of eating to manage weight [31]. An example question for the EmE subscale includes “When I feel anxious, I find myself eating”; an example question for uncontrolled eating includes “Sometimes when I start eating, I just can’t seem to stop”; and an example question for cognitive restraint includes “I deliberately take small helpings as a means of controlling my weight” [30]. The questions for each subscale are calculated based on participant responses using a 4-point Likert scale; where higher scores on each subscale indicate higher levels of the respective eating behavior. For analysis, the summed responses for each subscale were converted into a 100-point scale ranging from 0–100 [31].

The Control of Eating Questionnaire (CEQ) was used to evaluate food craving severity [32]. The CEQ is a validated 21-item scale that asks participants to rate their individual food craving experiences over the last week. Specifically, the craving control (CC) subscale reflects the control one has over their food cravings. CC was measured using a digital visual analog scale, where each question is presented above a 100-point line with anchor words at each end. An example question for the CC subscale with anchor words parenthesized includes “How difficult has it been to resist any food cravings? (“Not difficult at all” to “Extremely difficult”) [32,33]. CC was calculated as the average of the subscale’s items with higher scores reflecting lower control over cravings.

### 2.8. Statistical Analyses

Using moderate effect sizes of f = 0.25 for ANCOVA (2 groups, 4 covariates) and f^2^ = 0.15 for multiple regression (5 independent variables), it was determined that 128 participants would yield ≥80% power for both analyses at an α = 0.05. Notably, all ANCOVA and multiple and logistic regression models described hereafter were adjusted for sex (combined sample only) and race, and for BMI and fat-free mass given their well-documented influence on eating behaviors and dietary intakes [34]. As mentioned in the introduction, the inclusion of extraneous covariates into regression models often results in overfitting errors that reduce the likelihood of replicating findings in external samples [18,19]. Thus, no additional demographic covariates were included into regression models, and age was not included due to the small age range of our sample. Data for the multiple regression models are presented as the standardized coefficients (β). All data was normally distributed based on Shapiro–Wilk and/or visual inspection of Q-Q plots and all variance inflation factors across multiple regression models were <5. Moreover, all scales demonstrated good internal consistency (all Cronbach’s α ≥ 0.73).

Participant characteristics were evaluated by sex using independent t-tests. Multiple regression analyses were initially performed to evaluate the associations between the CESD, TFEQ, CEQ, and ERQ subscales. Multiple regression was then used to evaluate the associations between the dietary intake variables retained following LASSO regression and each questionnaires’ corresponding subscales for the combined sample and by sex. Dietary intake variables retained using LASSO that also demonstrated significant associations with depressive symptoms, and emotional regulation, CC, or eating behaviors, underwent further mediation analyses (hypothesized model: depressive symptoms > emotional regulation, CC, and/or eating behaviors > dietary intakes) using the methods described by Baron and Kenny [35]. For this, and after observing significant associations between the DV and both the primary IV (i.e., depressive symptoms) and a mediating variable (MV), the primary IV and the MV were simultaneously entered into the regression model. If the association between the primary IV remained significant and the MV was not, the association between the primary IV and the DV is not mediated by the MV. However, if the MV remains significantly associated with the DV and the primary IV does not, the MV is interpreted as a significant mediator of the relationship observed between the primary IV and the DV. If an MV was determined to be a significant mediator, it was then treated as the primary IV to determine its association with the DV after the inclusion of other potential mediators and the primary IV (now treated as an MV) into the model.

Dietary variables retained using LASSO that demonstrated significant associations with depressive symptoms, specifically, were entered as IVs into logistic regression models to determine their associations with (i) MDD and (ii) the use of mental health medications—defined as being prescribed and currently taking medications to treat depression, anxiety, and/or mood disorders. Associations with these dietary factors were evaluated using odds ratios (OR) before and after (adjusted OR [OR_adj_]) the inclusion of associated subscale scores for depressive symptoms, emotional regulation, CC, and eating behaviors. It is important to note that because all predictor variables employed within this logistic regression model were continuous variables, ORs are interpreted as the increased odds of an event (i.e., MDD or use of mental health medications) per unit increase in the predictor variable. For example, if added sugar produces a statistically significant OR of 1.1 for MDD (a small effect when using discrete predictors), individuals consuming 12.5 tsp of added sugar (the recommended cutoff for added sugar consumption) would be 13.75 times more likely to have MDD than those consuming no added sugar. As such, large ranges for predictor variables (e.g., 0–100% for relative sugar intake or 0-infinte tsp eq. for added sugar intake) often result in small ORs that are both statistically significant and clinically meaningful [36]. Finally, ANCOVA was used to evaluate the differences in the intake of the retained dietary variables between MDD and mental health medication groups. Notably, no dietary intakes were retained for the prediction of depressive symptoms in males and thus, this was not evaluated in males. Statistical significance was accepted at *p* ≤ 0.050. All analyses with the exception of LASSO (SPSS version 29) were conducted using R.

## 3. Results

### 3.1. Participant Characteristics

Participant characteristics are presented in Table 1. The sample was 57% female (*n* = 86) and 43% male (*n* = 65). Additionally, the sample consisted of 33.1% White (40.7% F, 23.1% M); 23.8% Black (27.9% F, 18.5% M); 39.7% Asian (25.6% F, 58.5% M); and 3.3% White Hispanic (5.8% F, 0.0% M) participants. Females had significantly lower height, weight, BMI, fat-free mass, expressive suppression, energy intake, and relative protein intake, and significantly higher body fat percent, fat mass, and ratings of EmE when compared to males (all *p* ≤ 0.025). A total of 20 participants (13.2%) were prescribed at least one mental health medication which included selective serotonin reuptake inhibitors (*n* = 12), serotonin–norepinephrine reuptake inhibitors (*n* = 3), norepinephrine–dopamine reuptake inhibitor (*n* = 3), 2 tricyclic antidepressants (*n* = 2), anxiolytics (*n* = 4), and antipsychotics (*n* = 1). Finaly, 51 participants (33.8%) were classified as being at risk for MDD.

### 3.2. Depressive Symptoms, Eating Behavior, and Emotional Regulation

The associations between depressive symptoms, eating behavior, and emotional regulation are presented in Table 2. For the combined sample and females, depressive symptoms were positively associated with uncontrolled eating, EmE, CC, and cognitive reappraisal (all *p* ≤ 0.049). Depressive symptoms were also positively associated with uncontrolled eating (*p* = 0.001) and EmE (*p* = 0.009) in males. For all groups, uncontrolled eating was positively associated with EmE and CC (all *p* ≤ 0.004) and EmE was positively associated with CC (all *p* ≤ 0.001). Expressive suppression was associated with cognitive reappraisal for males (*p* < 0.001).

All eating behavior subscales were significantly associated with MDD in the combined sample (*p* ≤ 0.044), though cognitive reappraisal and expressive suppression were not significantly associated (both *p* ≥ 0.060). For females, all variables were significantly associated with MDD (*p* ≤ 0.011), apart from cognitive restraint and expressive suppression (both *p* ≥ 0.079). Uncontrolled eating (*p* = 0.012) and EmE (*p* = 0.004) were the only variables significantly associated with MDD in males. EmE was the only variable that was significantly associated with the use of mental health medication for the combined sample (*p* = 0.032) and females (*p* = 0.015).

### 3.3. Added Sugar Intake

Results of the multiple regression and mediation analyses for added sugar intake are illustrated in Figure 1. Added sugar intake was the only positive dietary variable retained using LASSO in the combined sample that was also positively associated with depressive symptoms (β = 0.17, *p* = 0.043). Added sugar intake was also retained and positively associated with EmE (β = 0.21, *p* = 0.024) and CC (β = 0.28, *p* < 0.001). Added sugar intake was retained and positively associated with depressive symptoms for females (β = 0.25; *p* = 0.017) and with craving control (β = 0.43; *p* < 0.001) for males, but not with any other subscale across groups.

After the inclusion of CC into the depressive symptom model (IVs: depression and CC scores; DV: added sugar intake) for the combined sample, CC (β = 0.25, *p* = 0.003), but not depressive symptoms (β = 0.12, *p* = 0.164), remained significantly associated with added sugar intake. The inclusion of EmE (IVs: depression and EmE scores) into the depressive symptom model revealed that neither depressive symptoms (β = 0.12, *p* = 0.172) nor EmE (β = 0.17, *p* = 0.093) were associated with added sugar intake. However, only CC was significantly associated with added sugar intake after its inclusion into the EmE model before (β = 0.24; *p* = 0.008) (IVs: EmE and CC scores) and after the inclusion of depressive symptoms (β = 0.23; *p* = 0.011) (IVs: depression, EmE, and CC scores).

Results of the ANCOVA comparing the differences in added sugar intake between the risk for MDD and mental health medication groups are presented in Figure 2A–D. Added sugar intake was significantly higher in the MDD group and in those using mental health medications for both the combined sample (Figure 2A,C, MDD: F = 5.09; *p* = 0.026; medication use: F = 7.26; *p* = 0.008) and females (MDD: F = 5.70; *p* = 0.019; medication use: F = 6.85; *p* = 0.011).

The association between added sugar intake, and risk for MDD and mental health medication use, are presented in Figure 3 for the combined sample and in Figure 4 for females. Individuals with higher added sugar intake (IV) had significantly greater odds of being at risk for MDD (combined: OR = 1.05, *p* = 0.016; female: OR = 1.09, *p* = 0.012) and using mental health medications (combined: OR = 1.09, *p* = 0.023; female: OR = 1.10, *p* = 0.037). However, after the inclusion of both CC and EmE (as IVs) into the combined and female models, EmE returned as the only significant variable in the combined sample, where individuals with higher EmE had significantly greater odds of being at risk for MDD (OR = 1.07; *p* = 0.002); while females with higher CC (OR = 1.04; *p* = 0.013) and added sugar intake (OR = 1.10; *p* = 0.014) had significantly greater odds of being at risk for MDD. Further, only added sugar (IV) was significantly associated with mental health medication use in the combined sample, where those with higher added sugar intake had significantly greater odds of mental health medication use (OR = 1.08; *p* = 0.041). No variable maintained a significant association with mental health medication use in females (all *p* ≥ 0.073).

### 3.4. Relative Sugar Intake

Results of the female specific multiple regression and mediation analyses for relative sugar intake in are illustrated in Figure 5. For females, relative sugar intake was the only other positive dietary variable retained using LASSO that was also positively associated with depressive symptoms (β = 0.22, *p* 0.045) and EmE (β = 0.34, *p* = 0.007). After the inclusion of EmE into the depressive symptom model (IVs: depression and EmE scores; DV: relative sugar intake), EmE (β = 0.29, *p* = 0.030), but not depressive symptoms (β = 0.14, *p* = 0.219), were significantly associated with relative sugar intake.

Results of the ANCOVA showed that relative sugar intake was significantly higher for those using mental health medication in the combined sample (Figure 2B; F = 8.42, *p* = 0.004) and females (F = 8.07, *p* = 0.006); though relative sugar intake was not significantly different between MDD groups (Figure 2D; combined sample: F = 1.56, *p* = 0.214; females: F = 3.77; *p* = 0.056).

Females with higher relative sugar intake (IV) had significantly greater odds of being at risk for MDD (Figure 4; OR: 1.08, *p* = 0.029), which was not observed in the combined sample (Figure 3; OR = 1.04, *p* = 0.157). However, EmE was the only variable that was significantly associated with MDD in females after the inclusion of all variables into the model, where females with higher EmE scores had significantly greater odds of being at risk for MDD (Figure 4; OR = 1.06, *p* = 0.040). Individuals with higher relative sugar intake had significantly higher odds of using mental health medication before (combined: OR = 1.09, *p* = 0.046; female: OR = 1.12, *p* = 0.033) but not after (combined: OR = 1.07; *p* = 0.121; females: OR = 1.10; *p* = 0.116) the inclusion of all associated variables into the overall models (all *p* ≥ 0.082).

## 4. Discussion

This study assessed the associations between diet and depressive symptoms, the risk of major depressive disorder, and mental health medication use in a multi-ethnic sample of young adults. In addition, this study sought to determine whether eating behaviors and emotional regulation mediated those relationships. We hypothesized that specific dietary components would be linked to depression, and that these associations would be mediated by components of emotional regulation. In support of our hypotheses, this study is the first, to our knowledge, to show that after the rigorously narrowing from >100 dietary factors, only added and relative sugar consumption were associated with depressive symptoms. Additionally, our study is the first (to our knowledge) to show that CC and EmE mediated the relationship between sugar consumption and depressive symptoms. Overall, the findings from our study are relevant to nutrition and mental health professionals and present novel questions regarding the relationship between diet and components of mental health.

While our findings support the broader hypothesis that specific dietary components would be linked to depression, we did not anticipate that sugar consumption would be the only remaining positive dietary factor robust to exclusion. While our results add to a growing body of literature disclosing the associations between sugar consumption and depression [37], numerous studies have demonstrated these same associations, albeit with alternative nutrients [13,14,15,16]. As such, it has been highly challenging to sort and identify the strongest links between dietary factors and depression in practice without implementing major dietary changes with inherently low success rates. Furthermore, it remains challenging to identify which dietary factor is most reflective of depressive symptoms for patients and providers alike. Thus, we sought to isolate the most influential variables by mathematically excluding unnecessary variables and evaluating those remaining. Our study is unique in that it is the first to use this approach to distinguish the associations between depression and sugar consumption amongst an abundance of dietary factors, many of which having previously-demonstrated associations. Given that are there are an immeasurable number of potential associations between dietary factors and depression, the fact that added and relative sugar consumption remained robust to exclusion should increase confidence in the results of prior investigations showing similar relationships. However, it should not be understated that sugar consumption is one of many factors (dietary or otherwise) linked to depression, and practitioners and researchers should continue to comprehensively evaluate this condition.

Studies have speculated several underlying mechanisms for the relationship between depression and dietary sugar intake. In the brain, these include the effects of sugar consumption on dysfunctional neurotransmitter synthesis [38], overstimulation of the hypothalamic–pituitary–adrenal axis [39], and reduced brain-derived neurotrophic factor [40]. Systemically, potential mechanisms include the roles of sugar consumption in systemic inflammation [41], insulin resistance [42], and maladaptive intestinal microbiomes [43]. Although many of these mechanisms have theoretical plausibility, they have been difficult to isolate in human models due to the complexities of depressive disorders, where it is difficult to distinguish the causality of dietary factors apart from other inter-related components more likely to manifest in an uncontrolled and complicated environment. So, rather than excessive sugar intake promoting physiological shifts that increase depression, our findings suggest that excessive sugar consumption is simply reflective of the dysregulated eating behaviors that exacerbate, or are exacerbated by, depressive symptoms. Examples of this behavioral mechanism can be observed in the link between substance use and depression [44], where substances are often used to cope with the negative emotions that manifest in depressive symptoms, or to satisfy the cravings for the substance of choice. Interestingly, sugar consumption has been shown to be as addictive as illicit substances and is similarly linked to depression [45]. As such, it is possible that, like substance use, sugar intake reflects the dysregulated emotions that contribute to depression, rather than maladaptive physiological alterations.

Collectively, these findings have clear implications for those experiencing depressive symptoms or MDD. Prior research has emphasized emotional regulation as one pathway through which negative emotions, such as those commonly experienced by people with MDD or depressive symptoms, are connected to poor dietary and eating habits [46]. Consistent with these findings, emotion regulation focused interventions have improved eating behaviors and overall body mass index [47]. Mindfulness-based interventions, which are typically used to help people regulate their emotions, produce notable improvements in more intentional eating behaviors [48]. However, mindfulness-based interventions only exert small improvements in actual dietary intake [48], likely because these protocols lack specific references to healthy eating behaviors. Therefore, interventions focused on improving emotion regulation skills, coupled with nutrition education, may mitigate negative dietary intakes among young adults experiencing depression, particularly young adult females.

### Strengths and Limitations

Our study had several notable strengths. First, we employed rigorous selection procedures for our variables to identify unique associations between dietary and psychological variables. We also choose not to oversaturate our models with covariates, which, in addition to the removal of multicollinearity from LASSO regression procedures, should increase the confidence that these findings would replicate in external multi-ethnic samples of young adults. Further, assessing depressive symptoms, as well as major depression risk and mental health medication use, allowed us to capture a more comprehensive picture of how sugar intake impacts mental health parameters. Our study also had several limitations. While the 24-h food recall provides objective dietary data, these data, and the data from our behavioral subscales, are self-report and cross-sectional in nature; thus, causality cannot be established. Moreover, because this study was part of a larger line of investigation, a single 24-h food recall was employed. However, reports suggest that this technique is well-aligned with one’s actual/usual dietary intake. While our sample was multi-ethnic, the racial/ethnic distribution of our sample was different between males and females, and Hispanic individuals were underrepresented. Additionally, one-proportion z-tests revealed that our male and female distributions did not differ from the national proportion, but our racial/ethnic distribution did not align with the national (i.e., US) nor state (MS) level proportions. Future studies should evaluate the associations in underrepresented groups, particularly in a more evenly distributed and representative sample. Finally, our study was conducted only in young adults; however, there has been a paucity of literature in this group, of whom are more likely to suffer from emotional dysregulation when compared to their middle-aged and older adult counterparts.

## 5. Conclusions

In conclusion, our study revealed that added and relative sugar consumption are distinctly associated with depressive symptoms, but that these associations are mediated by EmE and CC. While these findings better elucidate the link between sugar and depression in young adults, and highlight the role of emotional dysregulation, more research evaluating this relationship is warranted. Still, our findings generate novel questions regarding the link between diet and mental health conditions and may be relevant for nutrition and mental health professionals.

## Figures and Tables

**Figure 1 healthcare-12-01944-f001:**
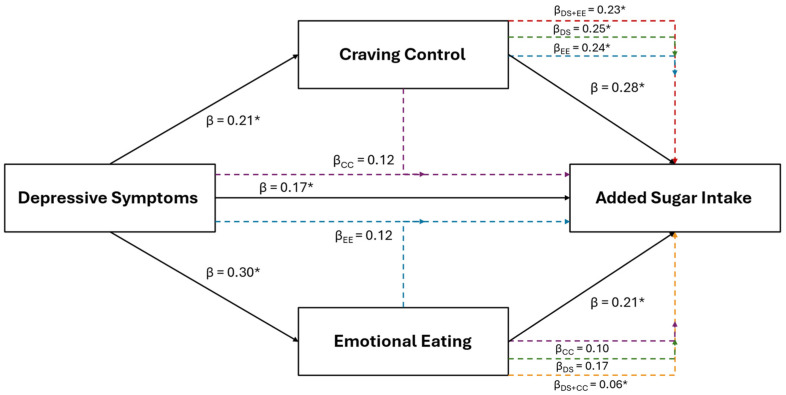
The relationship between depressive symptoms and added sugar consumption in the combined sample of young adults, and the mediating roles of craving control and emotional eating. The black solid lines represent independent associations between variables and the dashed intersecting lines represent the associations after adjusting for other variables presented in the figure. Purple dashed lines represent the association after adjusting for craving control; blue dashed lines represent the association after adjusting for emotional eating; green dashed lines represent the associations after adjusting for depressive symptoms; red dashed lines represent the associations after adjusting for depressive symptoms and emotional eating; and the orange dashed lines represent the associations after adjusting for depressive symptoms and craving control. Standardized β coefficients without subscripts represent the standardized coefficients of the independent associations (affiliated with the solid black lines), while standardized β coefficients with subscripts refer to the associations after adjusting for the mediators. * Ssignificantly associated at *p* < 0.050

**Figure 2 healthcare-12-01944-f002:**
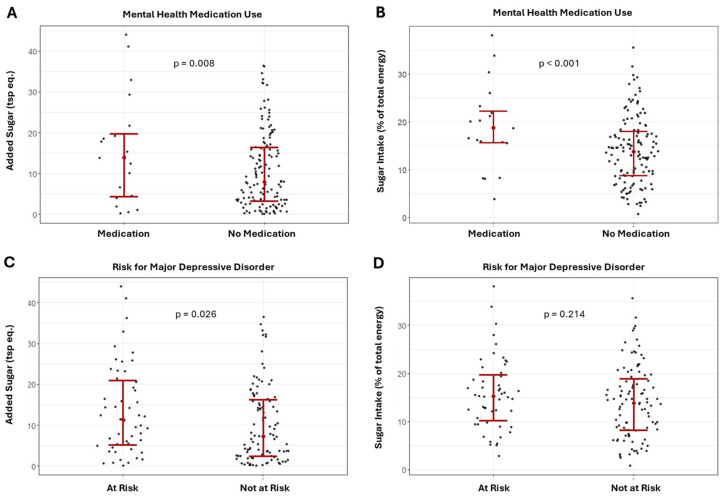
Differences in added and relative sugar intake between those using and not using mental health medications (**A**,**B**) and major depression risk groups (**C**,**D**) in the combined sample.

**Figure 3 healthcare-12-01944-f003:**
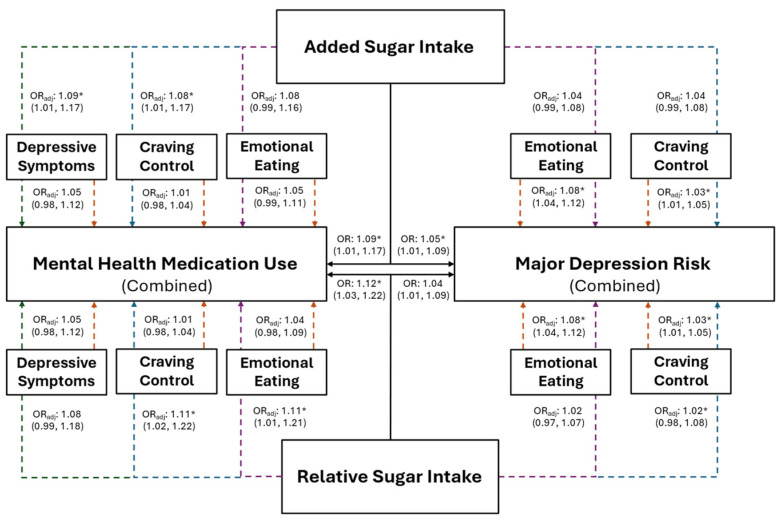
The predictive ability of added and relative sugar intake on the use of mental health medications and the risk for major depression in the combined sample of young adults. The black solid lines represent independent associations between added and relative sugar intake with mental health medication use and risk for major depression. The dashed lines represent the associations after adjusting for other variables presented in the figure. Orange dashed lines stemming from depressive symptoms, craving control, and emotional eating represent the associations between those variables after adjusting for added or relative sugar intake. The colored dashed lines stemming from added and relative sugar intake represent the associations between those variables and mental health medication use and major depression risk after adjusting for the variables they proceed through. Purple dashed lines represent the association after adjusting for emotional eating; blue dashed lines represent the association after adjusting for craving control; green dashed lines represent the associations after adjusting for depressive symptoms. Adjusted (OR_adj_) and unadjusted (OR) odds ratios are presented alongside their 95% confidence intervals. * Significantly associated at *p* ≤ 0.050.

**Figure 4 healthcare-12-01944-f004:**
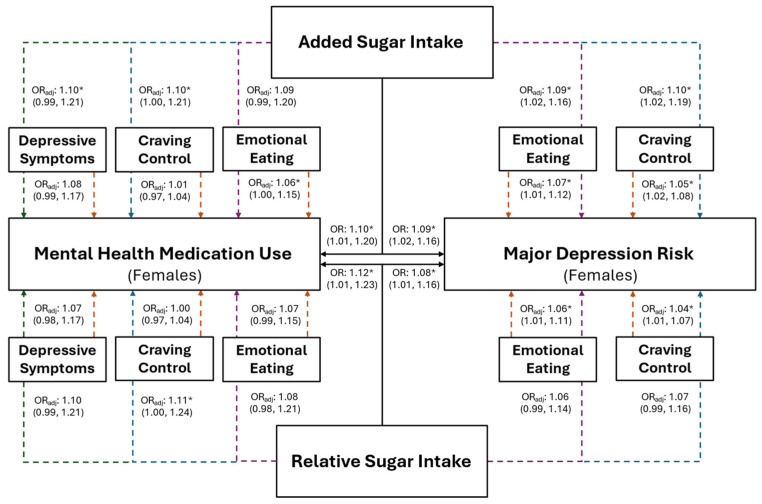
The predictive ability of added and relative sugar intake on the use of mental health medications and the risk for major depression in young adult females. The black solid lines represent independent associations between added and relative sugar intake with mental health medication use and risk for major depression. The dashed lines represent the associations after adjusting for other variables presented in the figure. Orange dashed lines stemming from depressive symptoms, craving control, and emotional eating represent the associations between those variables after adjusting for added or relative sugar intake. The colored dashed lines stemming from added and relative sugar intake represent the associations between those variables and mental health medication use and major depression risk after adjusting for the variables they proceed through. Purple dashed lines represent the association after adjusting for emotional eating; blue dashed lines represent the association after adjusting for craving control; and green dashed lines represent the associations after adjusting for depressive symptoms. Adjusted (OR_adj_) and unadjusted (OR) odds ratios are presented alongside their 95% confidence intervals. * Significantly associated at *p* ≤ 0.050.

**Figure 5 healthcare-12-01944-f005:**
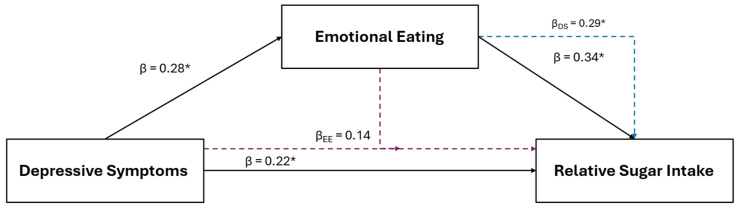
The relationship between depressive symptoms and relative sugar consumption in young adult females and the mediating role of emotional eating. The black solid lines represent independent associations between variables and the dashed intersecting lines represent the associations after adjusting for other variables presented in the figure. Purple dashed lines represent the association after adjusting for emotional eating; the blue dashed line represents the association after adjusting for depressive symptoms. Standardized β coefficients without subscripts represent the standardized coefficients of the independent associations (affiliated with the solid black lines), while standardized β coefficients with subscripts refer to the associations after adjusting for the mediators. * Significantly associated at *p* ≤ 0.050.

**Table 1 healthcare-12-01944-t001:** Participant Characteristics.

	Total (*n* = 151)	Females (*n* = 85)	Males (*n* = 65)
**Sex**			
Female	86 (57.0%)		
Male	65 (43.0%)		
**Race**			
White	50 (33.1%)	35 (40.7%)	15 (23.1%)
Black	36 (23.8%)	24 (27.9%)	12 (18.5%)
Asian	60 (39.7%)	22 (25.6%)	38 (58.5%)
Hispanic	5 (3.3%)	5 (5.8%)	0 (0.0%)
**Anthropometrics**			
Age (y)	21.4 ± 2.5	21.3 ± 1.9	21.4 ± 3.2
Height (cm)	167.7 ± 8.9	163.0 ± 6.5 *	173.9 ± 7.9
Weight (kg)	70.3 ± 17.1	66.8 ± 17.0 *	75.0 ± 16.2
BMI (kg/m^2^)	22.0 ± 5.1	21.2 ± 5.1 *	23.1 ± 4.9
Body fat (%)	29.6 ± 9.9	34.1 ± 8.7 *	23.5 ± 8.1
Fat mass (kg)	21.3 ± 11.4	23.8 ± 12.4 *	18.1 ± 9.1
Fat-free mass (kg)	49.0 ± 11.5	43.0 ± 7.1 *	56.9 ± 11.4
Mental Health Medication	20 (13.2%)	18 (20.9%)	2 (3.1%)
**CESD**			
Depressive symptoms	13.7 ± 8.1	14.0 ± 7.8	13.4 ± 8.5
Risk for Clinical Depression	51 (33.8%)	33 (38.4%)	18 (27.7%)
**TFEQ**			
Cognitive restraint	15.2 ± 11.3	14.9 ± 11.8	15.7 ± 10.8
Emotional eating	8.0 ± 11.5	10.5 ± 12.1 *	4.8 ± 9.8
Uncontrolled eating	10.6 ± 8.9	10.2 ± 9.0	11.0 ± 8.9
**CEQ**			
Craving control	43.0 ± 20.4	42.7 ± 20.5	43.3 ± 20.3
**ERQ**			
Cognitive reappraisal	28.9 ± 5.8	29.3 ± 14.7	28.3 ± 5.7
Expressive suppression	15.7 ± 5.0	14.7 ± 4.9 *	17.2 ± 4.9
**ASA24^®^**			
Total energy (kcals/d)	1974 ± 908	1807 ± 861 *	2193 ± 929
Protein (%)	17.9 ± 6.2	17.0 ± 5.3 *	19.2 ± 7.0
Fat (%)	38.4 ± 9.0	39.0 ± 8.6	37.6 ± 9.5
Saturated Fat (%)	12.0 ± 4.2	12.0 ± 4.1	12.1 ± 4.5
Carbohydrate (%)	44.4 ± 11.4	44.8 ± 10.6	43.9 ± 12.6
Fiber (%)	3.5 ± 1.9	3.7 ± 2.1	3.2 ± 1.7
Sugar (%)	14.7 ± 7.5	14.8 ± 7.3	14.5 ± 7.9
Added sugar (tsp eq.)	11.4 ± 10.0	10.8 ± 9.3	12.1 ± 10.9

Data are presented as mean ± standard deviation or as *n* (% of the column total). * significantly different from males at *p* ≤ 0.050. BMI: body mass index; CESD: Center for Epidemiologic Studies Depression Scale; TFEQ: Three Factor Eating Questionnaire 18-item; CEQ: Controlled Eating Questionnaire; ERQ: Emotional Regulation Questionnaire; ASA24^®^: Automated Self-Administered 24-Hour Dietary Assessment; tsp eq.: teaspoon equivalent.

**Table 2 healthcare-12-01944-t002:** Associations Between Depressive Symptom, Eating Behavior, and Emotional Regulation Scales.

	CESD	TFEQ			CEQ			ERQ
	Depressive Symptoms	Cognitive Restraint	Uncontrolled Eating	Emotional Eating	Craving Control	Craving Sweet	Craving Savory	Cognitive Reappraisal
**Combined Sample (*n* = 151)**								
Depressive Symptoms	-	-	-	-	-	-	-	-
Cognitive Restraint	0.16	-	-	-	-	-	-	-
Uncontrolled Eating	0.38 *	0.10	-	-	-	-	-	-
Emotional Eating	0.38 *	0.13	0.57 *	-	-	-	-	-
Craving Control	0.22 *	-0.03	0.52 *	0.38 *	-	-	-	-
Cognitive Reappraisal	−0.17 *	−0.03	0.09	0.11	0.10	0.09	0.04	-
Expressive Suppression	0.08	0.12	0.09	0.06	0.08	0.12	0.15	0.36
**Females**								
Depressive Symptoms	-	-	-	-	-	-	-	-
Cognitive Restraint	0.18	-	-	-	-	-	-	-
Uncontrolled Eating	0.38 *	0.13	-	-	-	-	-	-
Emotional Eating	0.39 *	0.22	0.57 *	-	-	-	-	-
Craving Control	0.32 *	0.00	0.47 *	0.35 *	-	-	-	-
Cognitive Reappraisal	−0.28 *	−0.14	0.13	0.08	0.14	−0.00	0.00	-
Expressive Suppression	0.06	0.05	0.09	0.01	0.20	0.18	0.24 *	0.22
**Males**								
Depressive Symptoms	-	-	-	-	-	-	-	-
Cognitive Restraint	0.14	-	-	-	-	-	-	
Uncontrolled Eating	0.40 *	0.04	-	-	-	-	-	-
Emotional Eating	0.34 *	0.00	0.49 *	-	-	-	-	-
Craving Control	0.06	−0.08	0.60 *	0.44 *	-	-	-	-
Cognitive Reappraisal	−0.03	0.14	0.02	0.17	0.08	0.19	0.12	-
Expressive Suppression	0.10	0.22	0.08	0.11	−0.07	0.00	0.03	0.50 *

Data are presented as the standardized beta coefficients. * association significant at *p* ≤ 0.050. CESD: Center for Epidemiological Studies Depression Scale; TFEQ: Three-Factor Eating Questionnaire; CEQ: Control of Eating Questionnaire; ERQ: Emotional Regulation Questionnaire.

## Data Availability

Data may be made available upon reasonable request to the corresponding author.

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
