# Peer review of "The Associations between Depression and Sugar Consumption Are Mediated by Emotional Eating and Craving Control in Multi-Ethnic Young Adults"

_healthcare, 2024, doi:10.3390/healthcare12191944_

Round 1
Reviewer 1 Report
Comments and Suggestions for Authors
This manuscript aimed to investigate the relationship between dietary habits and depressive symptoms. This manuscript is well-written and scientifically sounds. However, there are few minor comments from me on the manuscript.
1. The race composition was different between female and male. This maybe one of the confounding factors that affects the findings. Different races may have different genetic composition or eating habits which may affect the psychological aspects of the participants.
2. Since this is a cross-sectional study, the causal relationship between depressive symptoms and sugar intake was hard to determined. The authors should also try to conduct the mediation analysis by using added sugar and relative sugar intake as independent variables and depressive symptoms as dependent variables.
Author Response
Comment 1: This manuscript aimed to investigate the relationship between dietary habits and depressive symptoms. This manuscript is well-written and scientifically sounds. However, there are few minor comments from me on the manuscript.
Response 1: Thank you so much for your review of our manuscript and for your suggestions! We hope that we have addressed all of your concerns in the point-by-point responses presented hereafter.
Comment 2: The race composition was different between female and male. This maybe one of the confounding factors that affects the findings. Different races may have different genetic composition or eating habits which may affect the psychological aspects of the participants.
Response 2: Thank you so much for your comment. We have added the different racial composition of our male and female samples as a limitation of the study (lines 529-530).
Comment 3: Since this is a cross-sectional study, the causal relationship between depressive symptoms and sugar intake was hard to determine. The authors should also try to conduct the mediation analysis by using added sugar and relative sugar intake as independent variables and depressive symptoms as dependent variables.
Response 3: Thank you so much for your suggestion. Importantly, and as you correctly state, this is a cross-sectional study which does not allow for a causal relationship to be determined, irrespective of whether sugar intake variables are included as an independent or dependent variable (i.e., it simply shows an association, with the only difference being the what is placed on, for example, the x or y axis). In addition, the standardized coefficients and inferential statistics (i.e., p-value) produced for a pair of variables during simple multivariate (including model covariates) regression are mathematically equivalent irrespective of their assignment as an IV or DV (i.e., if an IV and DV are swapped the standardized coefficient and p-value are the same for the switched variables). For example, when added sugar is the DV and depressive symptoms are the IV, the standardized β = 0.17 and the p = 0.043. When added sugar is switched to the IV and depressive symptoms to the DV, the results remain the same at a standardized β = 0.17 and the p = 0.043. When mediators are included into multiple regression models, the p-values produced for a given IV and DV are mathematically equivalent irrespective of whether they are assigned to the IV or DV. For example, when added sugar is the DV, depressive symptoms is the IV, and craving control is the mediating variable (IV), the p = 0.164 for the association between added sugar and depressive symptoms and remains p = 0.164 when they are switched to either the IV or DV. Because the interpretation of the results are the same regardless of their position as the IV/DV, and because it is most likely that psychiatric components predict dietary intakes in this context, we believe that the analyses at present appropriately evaluate the data. The aforementioned rationale has been included in the manuscript (lines 169-188).
Reviewer 2 Report
Comments and Suggestions for Authors
In this paper, the authors used LASSO regression to identify dietary predictors of depression, and used the mediation analysis to examine the role of emotional eating/craving control in this pair of relationship. I think the topic is suitable for publication in Healthcare.
Some suggestions:
1. In the title: “depression and added and relative sugar consumption”. Is it possible for the authors to rephrase this? It really looks like a typo for new readers.
2. Line 26, p<=0.037. It might be helpful to provide the range of p-values instead of the upper bound.
3. It might be helpful for the authors to do an additional proof on this paper for grammar errors. For example, Line 52, “and individual” -> an individual; Line 108, “is ancillary too” -> is ancillary to.
4. Line 87: “psychometric components”. Did the authors mean “psychiatric components”?
5. Line 87: “the retained dietary intake”. It might be helpful to explain that they are the ones retained after LASSO.
6. Abstract says “5 hispanic white”, while Line 253 says “non-white hispanic”.
7. Lines 238-243 for the main statistical model. It would be helpful if the authors can provide more details on: (1) how their model was constructed; (2) which methods and statistical software were used to fit the models. It’s unlikely that the model in Figure 1 is the result of logistic regression. Did the authors run two mediation analyses for craving control and emotional eating each or combined them using a structural equation modeling approach?
8. Lines 334-340: These odds ratios are statistically significant, but really small. None of them exceeds 1.5. Is it possible for the authors to add more explanations on why these small effects are important? Are they on par with results from past studies?
9. I’m a little confused with the method. In the LASSO regression to identify relevant dietary items, the dietary items were used as independent variables to predict depression. The best predictors (e.g. added sugar consumption) were selected. However, in the main model (Figure 1), the predictor and outcome were flipped. Depression became the independent variable and dietary items became the dependent variable. Could the authors provide a rationale behind this choice? Because the mediation model was used, these two models (X -> M -> Y and Y -> M -> X) are not equivalent. It also renders the LASSO results less relevant as it did not seem to directly test the main hypothesis (depression predicting dietary consumption), but tested the reverse version of it.
Comments on the Quality of English LanguageIt might be helpful for the authors to do an additional proof on this paper for grammar errors. For example, Line 52, “and individual” -> an individual; Line 108, “is ancillary too” -> is ancillary to.
Author Response
Comment 1: In this paper, the authors used LASSO regression to identify dietary predictors of depression, and used the mediation analysis to examine the role of emotional eating/craving control in this pair of relationship. I think the topic is suitable for publication in Healthcare.
Response 1: Thank you so much for your review of our manuscript and for your suggestions! We hope that we have addressed all of your concerns in the point-by-point responses presented hereafter and believe that these suggestions have significantly improved the manuscript.
Comment 2: In the title: “depression and added and relative sugar consumption”. Is it possible for the authors to rephrase this? It really looks like a typo for new readers.
Response 2: Thank you so much for catching this! We have rephrased the title to “The associations between depression and sugar consumption are mediated by emotional eating and craving control in multi-ethnic young adults” so that it is clearer to new readers (line 1).
Comment 3: Line 26, p<=0.037. It might be helpful to provide the range of p-values instead of the upper bound.
Response 3: Thank you so much for your comment. We have added the range of p-values to this section as requested (line 27).
Comment 4: It might be helpful for the authors to do an additional proof on this paper for grammar errors. For example, Line 52, “and individual” -> an individual; Line 108, “is ancillary too” -> is ancillary to.
Response 4: Thank you so much for catching these errors! We have conducted additional proofs and have corrected any remaining grammatical errors.
Comment 5: Line 87: “psychometric components”. Did the authors mean “psychiatric components”?
Response 5: Thank you so much for catching this! We have corrected this term as requested (line 90).
Comment 6: Line 87: “the retained dietary intake”. It might be helpful to explain that they are the ones retained after LASSO.
Response 6: Thank you so much for your comment. We have rephrased this sentence so that it is clear that the retained dietary intakes are the intakes retained following LASSO regression procedures as requested (line 87).
Comment 7: Abstract says “5 hispanic white”, while Line 253 says “non-white hispanic”.
Response 7: Thank you so much for catching this! We have altered the line in the abstract so that it matches the formats in subsequent sections (line 18).
Comment 8: Lines 238-243 for the main statistical model. It would be helpful if the authors can provide more details on: (1) how their model was constructed; (2) which methods and statistical software were used to fit the models. It’s unlikely that the model in Figure 1 is the result of logistic regression. Did the authors run two mediation analyses for craving control and emotional eating each or combined them using a structural equation modeling approach?
Response 8: Thank you so much for your comment. We have included the statistical software used to perform our analyses in lines 285-286. We have also included the methods used to construct the model as requested (lines 253-264).
Comment 9: Lines 334-340: These odds ratios are statistically significant, but really small. None of them exceeds 1.5. Is it possible for the authors to add more explanations on why these small effects are important? Are they on par with results from past studies?
Response 9: Thank you so much for your comment. We have included a section in the statistical analysis section (lines 271-281) that provides an explanation of both the statistical and clinical significance of the odds ratios observed in our study. For reference, this section states: “It is important to note that because all predictor variables employed within this logistic regression model were continuous variables, ORs are interpreted as the increased odds of an event (i.e., MDD or use of mental health medications) per unit increase in the predictor variable. For example, if added sugar produces a statistically significant OR of 1.1 for MDD (a small effect when using discrete predictors), individuals consuming 12.5 tsp of added sugar (the recommended cutoff for added sugar consumption) would be 13.75 times more likely to have MDD than those consuming no added sugar. As such, large ranges for predictor variables (e.g., 0-100% for relative sugar intake or 0-infinte tsp eq. for added sugar intake) often result in small ORs that are both statistically significant and clinically meaningful.” An additional reference is also provided detailing similar findings from prior works (PMID: 35380079).
Comment 10: I’m a little confused with the method. In the LASSO regression to identify relevant dietary items, the dietary items were used as independent variables to predict depression. The best predictors (e.g. added sugar consumption) were selected. However, in the main model (Figure 1), the predictor and outcome were flipped. Depression became the independent variable and dietary items became the dependent variable. Could the authors provide a rationale behind this choice? Because the mediation model was used, these two models (X -> M -> Y and Y -> M -> X) are not equivalent. It also renders the LASSO results less relevant as it did not seem to directly test the main hypothesis (depression predicting dietary consumption), but tested the reverse version of it.
Response 10: Thank you so much for your comment! We have provided the rationale for this in lines 169-188 of the manuscript. For reference, we have provided this rationale hereafter: “Although the dietary variables were included as predictor variables during LASSO, retained dietary variables were included as dependent variables for multiple regression analyses. For example, if total dairy intake was retained as a correlate of craving control in the combined sample following LASSO, craving control was entered as the independent variable (IV) in the subsequent multiple regression model to determine its association with the dependent variable (DV) total dairy; given that psychiatric variables likely predict dietary intakes in the context of this study as opposed to the alternative. Importantly, LASSO regression procedures do not produce inferential statistics (i.e., do not provide p-values). Instead, it is simply a systematic approach for the removal of variables that contribute to elevated multicollinearity during regression procedures. In addition, the standardized coefficients and inferential statistics (i.e., p-value) produced for a pair of variables during simple multivariate (including model covariates) regression are mathematically equivalent irrespective of their assignment as an IV or DV (i.e., if an IV and DV are swapped the standardized coefficient and p-value are the same for the switched variables). When mediators are included into multiple regression models, the p-values produced for a given IV and DV are mathematically equivalent irrespective of whether they are assigned to the IV or DV. As such, using the dietary intakes as the predictors (IV) for LASSO regression did not im-pact their associations/interpretations in subsequent multiple regression models. Moreover, this allowed us to also remove multicollinearity amongst these variables whilst also limiting the type I error rate expected to increase if each of the >100 dietary variables were included as a DV in their own multiple regression model.”
Reviewer 3 Report
Comments and Suggestions for Authors
In my view, the research is well thought out and well prepared. However, there are some small issues that could be improved, which I highlight below:
- Regarding the sampling, the type of sampling should be specified, it is not enough with ‘through in-person and online word of mouth’, the process, the type (snowball?), the criteria followed for the search of the participants, etc. should be specified.
- The representativeness of the sample in relation to the target population should be reflected (simple statistical calculation).
- Measuring instruments should be described together with their psychometric properties of their original validation and validity.
- Although there are some limitations. It would be appropriate to present them under a separate heading.
Author Response
Comment 1: In my view, the research is well thought out and well prepared. However, there are some small issues that could be improved, which I highlight below:
Response 1: Thank you so much for your review of our manuscript and for your suggestions! We hope that we have addressed all of your concerns in the point-by-point responses presented hereafter.
Comment 2: Regarding the sampling, the type of sampling should be specified, it is not enough with ‘through in-person and online word of mouth’, the process, the type (snowball?), the criteria followed for the search of the participants, etc. should be specified.
Response 2: Thank you so much for your recommendation. We have specified the sampling procedures used in the current study as requested (lines 104-105).
Comment 3: The representativeness of the sample in relation to the target population should be reflected (simple statistical calculation).
Response 3: Thank you so much for your suggestion. We have added the representativeness of the sample’s demographics, as obtained using one-proportion z-tests, to those at both the national (US) and state (Mississippi) level in the limitations section (lines 530-533).
Comment 4: Measuring instruments should be described together with their psychometric properties of their original validation and validity.
Response 4: Thank you so much for your suggestion. Unfortunately, we must disagree with the changes being requested. The psychometrics components of each measurement instrument used in this study were described in detail as they pertain to their original validation investigations (which are cited accordingly) in the original submission. We have included that these instruments are validated measures as requested, but because the psychiatric instruments used in this study have well-known validity (which we now state for each instrument), and because the psychometric properties requested are unique to the original investigation (and not our own), we believe that the inclusion of psychometric properties of prior works would 1) cause confusion for the reader, and 2) not meaningful add to/be repetitive to what is already provided for each instrument.
Comment 5: Although there are some limitations. It would be appropriate to present them under a separate heading.
Response 5: Thank you so much for your suggestion. We have added a new heading prior to this section as requested (line 515).